# Smooth muscle cells affect differential nanoparticle accumulation in disturbed blood flow-induced murine atherosclerosis

Hunter A. Miller[1]☉, Morgan A. Schake[2]☉, Badrul Alam Bony 🄳[1], Evan T. Curtis[1], Connor C. Gee[1], Ian S. McCue[2], Thomas J. Ripperda, Jr.[2], Yiannis S. Chatzizisis[3], Forrest M. Kievit[1]*, Ryan M. Pedrigi 🄳[2]*

1 Department of Biological Systems Engineering, University of Nebraska–Lincoln, Lincoln, NE, United States of America, 2 Department of Mechanical and Materials Engineering, University of Nebraska–Lincoln, Lincoln, NE, United States of America, 3 Cardiovascular Division, University of Nebraska Medical Center, Omaha, NE, United States of America

☉ These authors contributed equally to this work.
* rpedrigi@unl.edu (RMP); fkievit2@unl.edu (FMK)

**Data Availability Statement:** All relevant data are within the paper.

## Abstract

Atherosclerosis is a lipid-driven chronic inflammatory disease that leads to the formation of plaques in the inner lining of arteries. Plaques form over a range of phenotypes, the most severe of which is vulnerable to rupture and causes most of the clinically significant events. In this study, we evaluated the efficacy of nanoparticles (NPs) to differentiate between two plaque phenotypes based on accumulation kinetics in a mouse model of atherosclerosis. This model uses a perivascular cuff to induce two regions of disturbed wall shear stress (WSS) on the inner lining of the instrumented artery, low (upstream) and multidirectional (downstream), which, in turn, cause the development of an unstable and stable plaque phenotype, respectively. To evaluate the influence of each WSS condition, in addition to the final plaque phenotype, in determining NP uptake, mice were injected with NPs at intermediate and fully developed stages of plaque growth. The kinetics of artery wall uptake were assessed *in vivo* using dynamic contrast-enhanced magnetic resonance imaging. At the intermediate stage, there was no difference in NP uptake between the two WSS conditions, although both were different from the control arteries. At the fully-developed stage, however, NP uptake was reduced in plaques induced by low WSS, but not multidirectional WSS. Histological evaluation of plaques induced by low WSS revealed a significant inverse correlation between the presence of smooth muscle cells and NP accumulation, particularly at the plaque-lumen interface, which did not exist with other constituents (lipid and collagen) and was not present in plaques induced by multidirectional WSS. These findings demonstrate that NP accumulation can be used to differentiate between unstable and stable murine atherosclerosis, but accumulation kinetics are not directly influenced by the WSS condition. This tool could be used as a diagnostic to evaluate the efficacy of experimental therapeutics for atherosclerosis.

**Funding:** We acknowledge support from an Institutional Development Award (IDeA) from the National Institute of General Medical Sciences (NIGMS) of the National Institutes of Health (NIH) to RMP and FK (P30GM127200) and the Nebraska Center for Integrated Biomolecular Communication to RMP (NIH NIGMS grant number P20GM113126), grant funding from the American Heart Association (AHA) to RMP (19CDA34660218), the National Institute of Biomedical Imaging and Bioengineering (NIBIB) of the NIH to RMP (R21EB028960), and from the National Heart, Lung, and Blood Institute (NHLBI) of the NIH to YC (R01HL144690). This material is based upon work supported by the National Science Foundation Graduate Research Fellowship under Grant No. DGE-1610400 to HM. The research was performed in part in the Nebraska Nanoscale Facility: National Nanotechnology Coordinated Infrastructure and the Nebraska Center for Materials and Nanoscience, which are supported by the National Science Foundation under Award ECCS: 2025298, and the Nebraska Research Initiative. The funders had no role in study design, data collection and analysis, decision to publish, or preparation of the manuscript.

**Competing interests:** The authors have declared that no competing interests exist.

## Introduction

Atherosclerosis is a chronic inflammatory disease characterized by the development of plaques composed of lipids and immune cells within the artery wall. It is the leading cause of death worldwide [1]. The clinical significance of the plaque is typically based on the degree of lumen stenosis determined from angiography. However, the plaque phenotype that causes most deaths due to vulnerability to rupture does not always cause a severe stenosis [2]. While other imaging modalities can be used to better quantify additional features of plaque phenotype (e.g., intravascular ultrasound), all of them require invasive catheterization that carries risk of major complications to the patient [3]. Another problem with diagnosis of high-risk plaques is that progression to rupture is a highly nonlinear process, where plaque features can change rapidly in the weeks to months before rupture [4, 5]. Thus, there is a need for a noninvasive diagnostic tool that would allow more frequent imaging of patients considered intermediate-to-high risk for myocardial infarction or ischemic stroke [5]. Given the dynamic nature of plaque evolution, such a tool could also be impactful in research that employs animal models to characterize plaque progression and regression. To address this need, contrast-enhancing nanoparticles (NPs) can be tracked in atherosclerotic arteries to relate accumulation in plaques to plaque phenotype and changes with progression. While studies have examined NP accumulation in atherosclerosis [6], in general, few have looked at changes with plaque progression and the large range of NP properties alone, including size, targeting ligands, and surface coating, warrant additional studies.

Another factor that might influence NP accumulation is the blood flow environment, but no study to our knowledge has examined this relationship. In addition, blood flow is an important determinant of the susceptibility of an arterial segment to atherosclerosis [7–10]. This connection stems from the tangential load that blood flow imparts onto the endothelium called wall shear stress (WSS). In straight arterial segments, blood flow is laminar and the associated WSS promotes normal endothelial cell functions that are atheroprotective. In contrast, arterial segments around bifurcations or the inner curvature of highly curving arteries cause blood flow to be disturbed and the associated WSS is low in magnitude and/or varying in direction over the cardiac cycle (i.e., multidirectional), which causes an atherogenic endothelial phenotype [11–13]. WSS is not only important in atherosclerosis initiation, but also progression, where different WSS conditions have been shown to influence the development of different plaque phenotypes [8, 9, 14, 15].

We have previously reported that implantation of blood flow-modifying stents within the coronary arteries of transgenic hypercholesterolemic pigs caused the development of advanced plaques, wherein low WSS promoted thin cap fibroatheroma and multidirectional WSS was associated with the development of thin and thick cap fibroatheroma and pathological intimal thickening [1]. This study was motivated by a well-established ApoE$^{-/-}$ mouse model that places a blood flow-modifying constrictive cuff around one of the carotid arteries to induce disturbed flow and cause the development of advanced plaques [10, 16]. Recently, we quantified the WSS characteristics in the arteries of ApoE$^{-/-}$ mice instrumented with a cuff *in vivo* using micro-CT imaging and Doppler ultrasound to obtain each cuffed artery geometry and blood velocity, respectively, and then performed computational fluid dynamics to compute the associated WSS [11]. We found that the cuff reproducibly induced low WSS in the upstream region, high WSS within the cuff, and multidirectional (including oscillatory) WSS in the downstream region of each instrumented artery. These results aligned with those previously reported by the seminal study introducing this mouse model, which used an idealized CFD model to estimate WSS in the instrumented artery [10]. This study also showed that 9 weeks after cuff placement, upstream plaques exhibited increased lipids and inflammatory mediators,

together with decreased smooth muscle cells and collagen that localized to the cap region, similar to an unstable advanced plaque phenotype in humans, whereas downstream plaques showed the opposite trends and were more similar to a stable advanced plaque phenotype [10]. These results have been corroborated by several other studies [16–19].

Herein, we employed this well-established ApoE[-/-] mouse model to characterize differential accumulation of folic acid-coated gadolinium (FA-Gd) NPs between the two atherogenic WSS conditions induced by the cuff, low and multidirectional. Folic acid was chosen for the surface coating to improve colloidal stability, improve magnetic resonance imaging (MRI) enhancing properties, and take advantage of the folate receptor on activated macrophages that accumulate at the plaque site [20]. The kinetics of NP uptake into the plaques were determined *in vivo* using dynamic contrast-enhanced (DCE)-MRI. To better separate the influence of WSS versus plaque phenotype on NP accumulation, mice were injected with NPs and imaged at both intermediate (5 weeks) and fully-developed (9 weeks) stages of plaque growth. At the endpoint of the study, histology was performed to characterize plaque features, which were individually correlated to NP accumulation.

## Materials and methods

### Synthesis of FA-Gd NPs

FA-Gd NPs were formulated using a modified polyol synthesis method [21]. The NPs were formed by adding 2 mmol gadolinium chloride hydrate ($GdCl_3xH_2O$) into 30 mL of triethylene glycol and heated to 80˚C until the dissolution of precursors was achieved. The mixture was combined with 6 mmol sodium hydroxide and continuously stirred for 4 h at 180˚C. The NPs were coated with 4 mmol folic acid ($C_{19}H_{19}N_7O_6$) and continuously stirred for 12 h at 150˚C. Once cooled, the synthesized NPs were washed three times in deionized water from a Millipore water purification system. All chemicals were purchased from Sigma-Aldrich.

### NP relaxivity characterization

$R_1$ relaxivity of NPs was measured using a 9.4T (400 MHz) 89 mm vertical bore magnet (Varian, Walnut Creek, CA) with a 4 cm Millipede RF imaging probe and triple axis gradients (100 G/cm max). A fast spin echo sequence was used with the following parameters: 7 repetition time (TR) values from 200–2000 ms in 300 ms increments, echo time of 32.42 ms (TE), echo train length (ETL) of 16, 25x25x3 mm$^3$ field of view (FOV), and a 128x128 data matrix. The saturation recovery method was utilized to measure relaxation time, $T_1$, of each NP concentration based on MR signal and the following equation:

$$S = S_0 \left(1 - e^{-\frac{TR}{T_1}}\right) \tag{1}$$

In which S is MR signal at a given voxel and $S_0$ is the signal of the given voxel at saturation. Relaxivity, r in $mM^{-1}s^{-1}$, was then calculated as:

$$R_1 = r * C + b \tag{2}$$

Where $R_1 = T_1^{-1}$, C is the concentration of NP and b is the y-intercept of the line.

### Mouse model

This study was carried out in strict accordance with the recommendations in the Guide for the Care and Use of Laboratory Animals of the National Institutes of Health. The protocol was approved by the Institutional Animal Care and Use Committee (IACUC) of the University of

Nebraska-Lincoln (Project ID: 1581 and 2007). All surgeries were performed under isoflurane gas anesthesia (2–3% induction and 0.25–2% for maintenance) and buprenorphine (1 mg/kg via subcutaneous injection) was given at the time of surgery to provide sustained release of analgesia for 72 hrs. A total of 18 female ApoE$^{-/-}$ mice at 11 weeks of age were placed on an atherogenic diet (Envigo) for two weeks and then instrumented with a blood flow-modifying cuff (Promolding) around the left carotid artery that tapers from 500 μm at the inlet to 250 μm at the outlet. This cuff induces three regions of disturbed flow and associated WSS: low WSS upstream, high WSS within, and multidirectional WSS downstream of the cuff [11]. Previous studies have demonstrated that atherogenic flow conditions cause development of advanced plaques consistent with an unstable phenotype in the upstream region and a stable phenotype in the downstream region (no plaque forms in the cuff as high flow is atheroprotective) within 9 weeks of cuff placement [10, 16]. To evaluate NP accumulation at an intermediate and the fully-developed stage of plaque growth, five of these mice were injected with FA-Gd NPs and imaged with DCE-MRI at 5 and 9 weeks after cuff placement. Healthy 6-week-old C57BL/6 mice (four total arteries obtained from two mice) on a normal chow diet were also injected with NPs and imaged at a single time point as an additional control. The remaining ApoE$^{-/-}$ mice, in addition to those injected with NPs, were used for histological evaluation of plaque size and constituents at 5 versus 9 weeks and upstream versus downstream plaques.

## DCE-MRI and mouse imaging protocol

DCE-MRI was performed on mice herein as previously described [21, 22]. Briefly, mice were induced with 2% isoflurane gas and breathing rates were monitored by a pressure-based sensor (SA Instruments), maintaining 50 to 80 breaths per minute over the course of the imaging sequence. Mice were affixed in a cylindrical animal holder to maintain head and body position during MR imaging. A 30-G needle was connected to a syringe via catheter and inserted into a lateral tail vein and secured with surgical tape. Mice were injected with 100 μL FA-Gd NP solution followed by a 100 μL bolus of PBS to flush excess NPs from the catheter. DCE-MRI was performed using a 2-D gradient echo multi-slice sequence with two flip angles for calculation of $R_1$ [22] and generation of $K^{trans}$ maps, which indicates the NP permeation from the plasma into the arterial wall, based on the Patlak model, a two-compartment pharmacokinetic model assuming unidirectional contrast transfer. Two baseline scans were performed to calculate $T_1$ values prior to injection with the following parameters: TR = 138 ms, TE = 3.48 ms, flip angles of 10˚ and 30˚ respectively, 2 averages, 256x256 data matrix, 20 slices each with a 23x23x0.5 mm$^3$ FOV for a total scan time of 53 s. Post-contrast scans all used a flip angle of 30˚ with the same parameters as above. After completion of baseline scans, NPs were injected through the tail vein catheter, followed by 45 minutes of post-contrast image collection. The $K^{trans}$ maps were then generated using a custom program in MATLAB (R2020b). Pharmacokinetic properties of the NPs were determined based on a modification of our previously described methods, using the Patlak model instead of the reference region [22]. Briefly, arterial input functions were collected from ROIs typically drawn around the carotid arteries and fit to a bi-exponential model using a nonlinear least squares algorithm in MATLAB.

## Histological analysis

To assess the development of atherosclerotic plaques at 5 ($n$ = 8 mice) and 9 ($n$ = 10 mice) weeks after cuff placement, the carotid arteries of all mice were prepared for histological evaluation. Mice were intracardially perfused with 35 ml of saline solution and then perfusion-fixed with 35 ml of 4% paraformaldehyde at mean arterial pressure. Both carotid arteries were excised still attached to the aortic arch and embedded in OCT. The tissue was then snap frozen

in a mixture of isopentane (ThermoFisher) and dry ice. Cryosections with a thickness of 8 μm were serially collected from the innominate bifurcation in the right (control) common carotid artery, which aligned with a similar position in the far upstream region of the adjacent left (instrumented) common carotid artery (2–3 mm from the aortic arch), to the carotid bifurcation. Sections were collected in a way to allow assessment of multiple stains over the length of the arteries. Co-registration of the histology sections to the DCE-MRI images and associated $K^{trans}$ maps was done using the cuff as a landmark, which was easily identifiable in both MRI and histology, the latter because the cuff region remains non-diseased due to the presence of high WSS—plaques develop immediately upstream and downstream of the cuff in the atherogenic flow regions of the instrumented artery (Fig 1). Four plaque constituents were evaluated. Oil red O (Sigma) was performed for analysis of lipid and picrosirius red (Sigma) for analysis of collagen. These basic stains were imaged at 10x magnification with a Zeiss Axio Observer 5 microscope. Two immunostains were also performed to evaluate the presence of smooth muscle cells and mitotic cells, respectively. Smooth muscle cells were detected using a primary antibody directed against alpha-smooth muscle actin (α-SMA) (rabbit polyclonal anti-αSMA at 1:250, Abcam #ab5694) and an Alexa Fluor 594 preadsorbed secondary antibody (goat polyclonal anti-rabbit at 1:1000, Abcam #ab150084). Mitotic cells were detected using a primary antibody directed against Ki67 (rabbit polyclonal anti-Ki67 at 1:100, Abcam #ab15580) and an Alexa Fluor 647 secondary antibody (goat polyclonal to rabbit at 1:750, Abcam #ab150079). Both immunostains were counterstained with DAPI (0.0025%, Abcam #ab228549). Immunostained-sections were imaged at 20x magnification with a confocal microscope. Confocal parameters were held constant over all sections of all mice.

Sections stained for oil red O, picrosirius red, and Ki67 were binarized with a threshold in ImageJ to quantify the presence of positive staining within the plaque. The fluorescent images of α-SMA were converted to greyscale to quantify stain intensity. Quantification of all stains was done using a custom MATLAB program that summed the binarized (0 or 1) or greyscale (0 to 255) pixel values and normalized by the total number of pixels in the region of interest within the histological section (grey scale pixel values were additionally normalized by the maximum pixel value of 255). This program was also used to manually segment the arterial layers (lumen, internal elastic lamina (IEL), and external elastic lamina (EEL)) of each histological section (for fluorescence images, a phase image was taken with each that was used for segmentation). Once manually segmented, the program automatically identified the plaque based on deviations between the lumen and IEL contours (these contours overlap in a non-diseased vessel section). The "cap region" was also automatically identified as the first 13.3 μm of thickness from the plaque-lumen interface, which scales to ~150 μm in a human carotid artery —a typical thickness for the cap of a fibrous cap atheroma [23]. The program quantified stain area in the plaque, plaque cap, and plaque body minus the cap region (the phase images associated with the α-SMA stain of each section were used for quantification of plaque area and burden). Finally, the program also quantified the mean distribution of collagen and α-SMA within the plaque of each section from the IEL to the lumen.

## Statistics

To evaluate differences in NP accumulation between the different WSS regions, $K^{trans}$ was averaged over the three DCE-MRI slices nearest the cuff in each of the two regions of the instrumented artery where atherosclerosis develops, upstream (low WSS) and downstream (multidirectional WSS), from each mouse at 5 and 9 weeks after cuff placement. The contralateral control arteries from these mice were similarly evaluated, except $K^{trans}$ was averaged over the entire central portion of the artery. To evaluate differences in plaque constituents and size

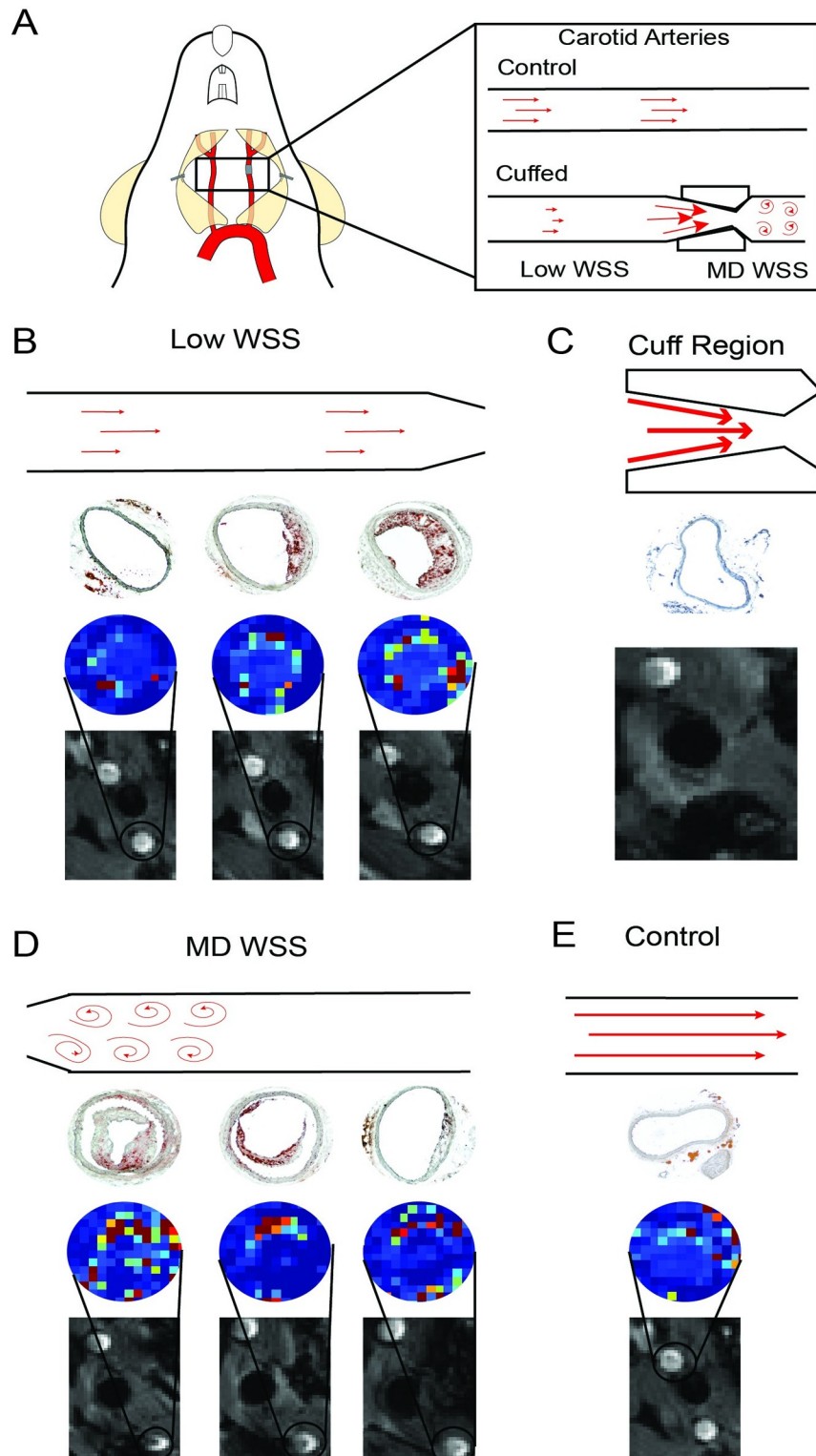

**Fig 1. Representative sequence of co-registered MR images and histological data along the length of the instrumented carotid artery.** (A) Diagram of the flow environment in the instrumented versus contralateral control carotid arteries. The cuff creates three distinct hemodynamic conditions, low WSS upstream of the cuff, high WSS within the cuff, and multidirectional (MD) WSS downstream of the cuff [11]. (B-D) Longitudinal sequence of DCE-MR images and K$^{trans}$ maps co-registered to histology sections stained with oil red O to identify lipid from the

(B) low WSS, (C) cuff, and (D) multidirectional WSS regions in the instrumented artery of a representative mouse at 9 weeks after cuff placement. (E) Data from the contralateral control artery. Note, the cuff prevents reliable acquisition of $K^{trans}$ maps, so they are not shown in this region.

between the different WSS regions and time points, each histological readout was averaged over all viable sections within the plaque of each instrumented vessel segment of each mouse. Group comparisons were performed using a one-way ANOVA. Pairwise comparisons, including those performed post hoc of the ANOVA, were done using a two-sample, two-tailed $t$-test. Comparisons were chosen a priori. A step-down Bonferonni-Holm correction method was used to control for type-I errors associated with multiple comparisons. Assumptions on normality were met based on a Shapiro-Wilk test and evaluation of histograms.

To evaluate the relationship between $K^{trans}$ and plaque area, lipid, collagen, and smooth muscle, respectively, one to three co-registered histology-MRI pairs were obtained in each of the two WSS regions, low and multidirectional (all mice are represented in this analysis for each vessel segment, but, in vessel segments of some mice, missing or damaged histology sections allowed only one pairing to be included in the analysis). Each histological stain was averaged over all sections associated with the co-registered MRI slice (typically, four histology sections per slice) and plotted as a function of $K^{trans}$. A Spearman's rank correlation coefficient ($\rho$) was calculated for each data set and a linear regression was performed to visualize the trend. A Bonferonni correction method was used with these data to account for multiple comparisons.

All statistical tests were performed in MATLAB. Quantities are reported as mean ± standard deviation (SD). Corrections for multiple comparisons were implemented as an adjustment of the calculated $p$-value by the ratio of 0.05 to the adjusted alpha value. An adjusted $p$-value of less than 0.05 was considered statistically significant, which is indicated as $^{*}p<0.05$, $^{**}p<0.01$, and $^{***}p<0.001$.

## Results

### Validation of NP structure and properties

Gd-based small molecule contrast agents (CAs) are commonly used in clinical settings to evaluate vascular permeability with DCE-MRI [6]. Concerns over toxicity and mediocre contrast enhancement have inspired the development of paramagnetic NP-based CAs often utilizing iron oxide or lanthanide series ions, like Gd [24, 25]. However, for these NPs to be useful in DCE-MRI, they must provide sufficient $T_1$ contrast enhancing properties. The FA coating, which was verified by FT-IR, functions as a surface coating [26] and serves to ensure strong $T_1$-enhancement, target engagement, and stability in blood. Surface FA preferentially targets folate receptors present on mononuclear phagocytes [1], increasing NP concentration at sites of inflammation by translocation [1, 27]. Surface FA also increases colloidal stability of FA-Gd, drawing $H_2O$ nearer to the core by increasing hydrophilicity and further increasing $T_1$ effects [28, 29].

FA-Gd NP dimensions were assessed using high-resolution transmission electron microscopy (HRTEM), which demonstrated a mean core diameter of 4.5 nm (**Fig 2A**). Hydrodynamic size was determined as 12.9±0.4 nm by dynamic light scattering (DLS). FT-IR was performed to confirm the presence of FA on the surface of FA-Gd NPs (**Fig 2B**). Shifting of the COOH peak from the free FA spectrum indicated coating of FA on the Gd NP surface. Prior to DCE-MRI, the contrast-enhancing properties of FA-Gd were also quantified. Relaxivity characterization of FA-Gd used a saturation recovery method of $T_1$ measurement at 9.4 T

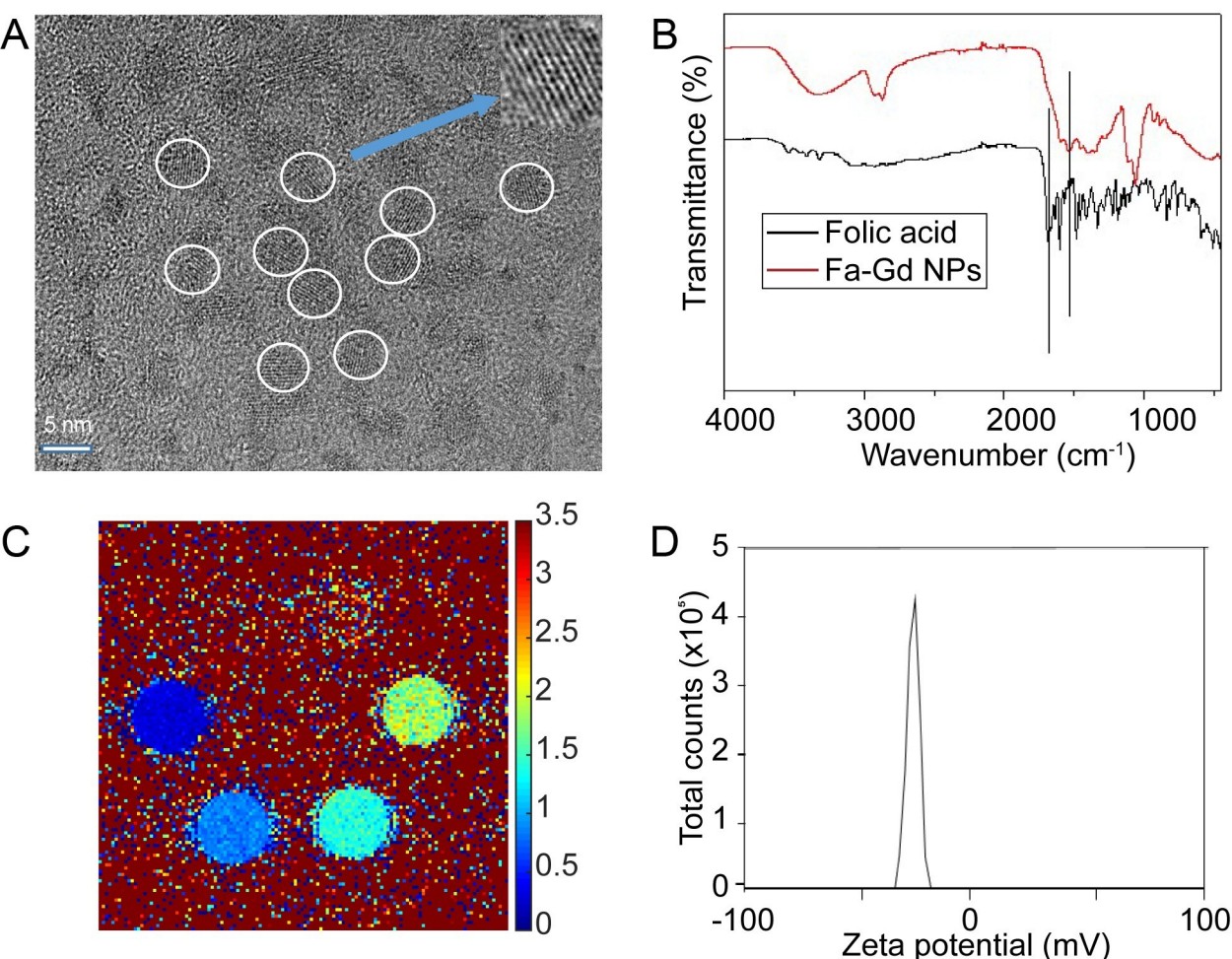

**Fig 2. NP structure and properties.** (A) HRTEM images of FA-Gd NPs, white circles indicate average diameter (4.5 nm) of the NPs. Blue arrow indicates lattice fringe of NPs. (B) FT-IR spectra of folic acid and FA-Gd NPs. The black spectrum corresponds to free FA, while the red spectrum corresponds to FA-Gd NPs. The COOH peak was shifted from 1680 to 1527 $cm^{-1}$, which confirmed the FA coating on to the Gd NPs. (C) $R_1$ map of FA-Gd NP phantoms for determination of $R_1$ relaxivity (3.14 $mM^{-1}s^{-1}$). (D) Zeta potential distribution of FA-Gd NPs, mean value of -40.5 mV.

and determined an $R_1$ relaxivity of 3.15 $s^{-1}mM^{-1}$ (**Fig 2C**). Zeta potential assessment of FA-Gd showed a mean value of -40.5 mV (**Fig 2D**), which increased colloidal stability and helped to prevent aggregation and adsorption of negatively-charged serum proteins [30].

## NP accumulation in arterial segments exposed to low versus multidirectional WSS

DCE-MRI provides the ability to noninvasively monitor accumulation and retention kinetics of a contrast agent into a tissue of interest and is typically used to measure blood vessel permeability. We have recently extended the use of DCE-MRI to assess and compare vascular permeability kinetics of different NPs in a mouse model of traumatic brain injury [21, 22]. Here, we used this same imaging technique to determine accumulation kinetics of FA-coated, Gd-core NPs with a mean core diameter of 4.5 nm in atherosclerotic plaques. A small core size served to increase $T_1$ effects via increased surface area-to-volume ratio and the resultant increase in Gd-$H_2O$ interactions [31], as well as minimize $T_2$ effects by decreasing the disruption of $H_2O$ proton spin phase coherence [32]. Though the core size of FA-Gd is below 5 nm, the

hydrodynamic size of 12.9 nm is sufficiently large to limit renal clearance [33]. This minimization of renal clearance, and thus increased duration of higher NP concentrations in the blood, may increase uptake into plaques, though previous investigations have shown plaque-associated macrophage internalization of NPs begins within 15 minutes of administration [34]. The relatively rapid NP clearance, compared with long-circulating formulations whose half-lives can be in the realm of hours and days [35], reduces exposure to off-target tissues. The size of FA-Gd is sufficiently small to limit sequestration to the liver by the reticuloendothelial system and increase the time spent in circulation [36].

Behavior of NPs in the blood (**Table 1**) was assessed by quantifying half-lives. The distribution half-life ($t_{1/2,dist}$) was 17.4±10.3 minutes while the elimination half-life ($t_{1/2,elim}$) was 46.8±14.5 minutes. As a comparison, the small molecule contrast agent Gd-DTPA has a $t_{1/2,dist}$ of approximately 4.0 minutes and a $t_{1/2,elim}$ of approximately 15.7 minutes [22]. We used DCE-MRI to generate $K^{trans}$ maps to determine the accumulation kinetics of NPs in different regions of the instrumented artery, which contain different WSS conditions, at intermediate and fully-developed stages of plaque growth. T1-weighted images and Gd-time curves were also generated (**S1 and S2 Figs**), though $K^{trans}$ was the metric of comparison for NP accumulation kinetics throughout the work. In both regions, plaques developed immediately adjacent to the cuff, had a length of approximately 0.5–1.5 mm, and exhibited the highest $K^{trans}$ in the portion of each region with maximum plaque burden (**Fig 3**). In the low WSS region, $K^{trans}$ substantially increased from the least diseased portion of the instrumented artery (three MRI slices from the cuff) to the portion containing the maximum plaque burden immediately adjacent to the cuff by 1.5-fold at 5 weeks and 2.2-fold at 9 weeks after cuff placement. In the multidirectional WSS region, no differences in $K^{trans}$ were seen, likely because this region abuts the carotid bifurcation, which contains naturally-occurring disturbed flow, so plaques were present throughout (**Fig 3**).

We next evaluated differences in maximal $K^{trans}$ between low and multidirectional WSS in the instrumented arteries, and normal WSS in the control arteries at 5 and 9 weeks after cuff placement. A one-way ANOVA demonstrated a highly significant difference between two or more of these groups (***$p<0.001$) and a two-sample, two-tailed $t$-test was used to evaluate pairwise differences (**Fig 4**). At the intermediate stage of plaque development (5 weeks), DCE-MRI showed a significant difference in NP accumulation between the low WSS region of the instrumented vessels compared to the contralateral control vessels, but not the multidirectional WSS region. The low WSS region had a mean $K^{trans}$ value across all mice of 0.055 ±0.0047 $min^{-1}$ versus 0.038±0.0053 $min^{-1}$ in the control (a 1.45-fold increase; **$p = 0.005$) and the multidirectional WSS region had a mean $K^{trans}$ value of 0.055±0.016 $min^{-1}$ versus 0.038 ±0.0053 $min^{-1}$ in the control (a 1.45-fold increase; $p = 0.22$). At the fully developed stage of plaque growth (9 weeks), the $K^{trans}$ difference between instrumented and control arteries was significant in the multidirectional WSS region, but not in the low WSS region. The low WSS region had a mean $K^{trans}$ value of 0.031±0.0067 $min^{-1}$ versus 0.025±0.0047 $min^{-1}$ in the control ($p = 0.38$), while the multidirectional WSS region had a mean $K^{trans}$ value of 0.050±0.011 $min^{-1}$ versus 0.025±0.0047 $min^{-1}$ in the control (a 2.00-fold increase; **$p = 0.007$). The low WSS region was also found to be statistically lower at 9 compared to 5 weeks (**$p = 0.0012$), whereas there was no difference between time points in the multidirectional WSS region ($p = 1$). Direct comparisons of mean $K^{trans}$ values in the low versus multidirectional WSS regions also demonstrated a significant difference at 9 weeks (1.64-fold; *$p = 0.043$), but not 5 weeks ($p = 1$).

**Table 1. Pharmacokinetics of FA-Gd in ApoE[-/-] mice determined by MRI.**

| Distribution Half-life ($t_{1/2,dist}$) | $K_{el,dist}$ [$min^{-1}$] | Elimination Half-life ($t_{1/2,elim}$) | $K_{el,elim}$ [$min^{-1}$] |
|---|---|---|---|
| 17.36±10.26 | 0.067±0.068 | 46.82±14.50 | 0.016±0.0052 |

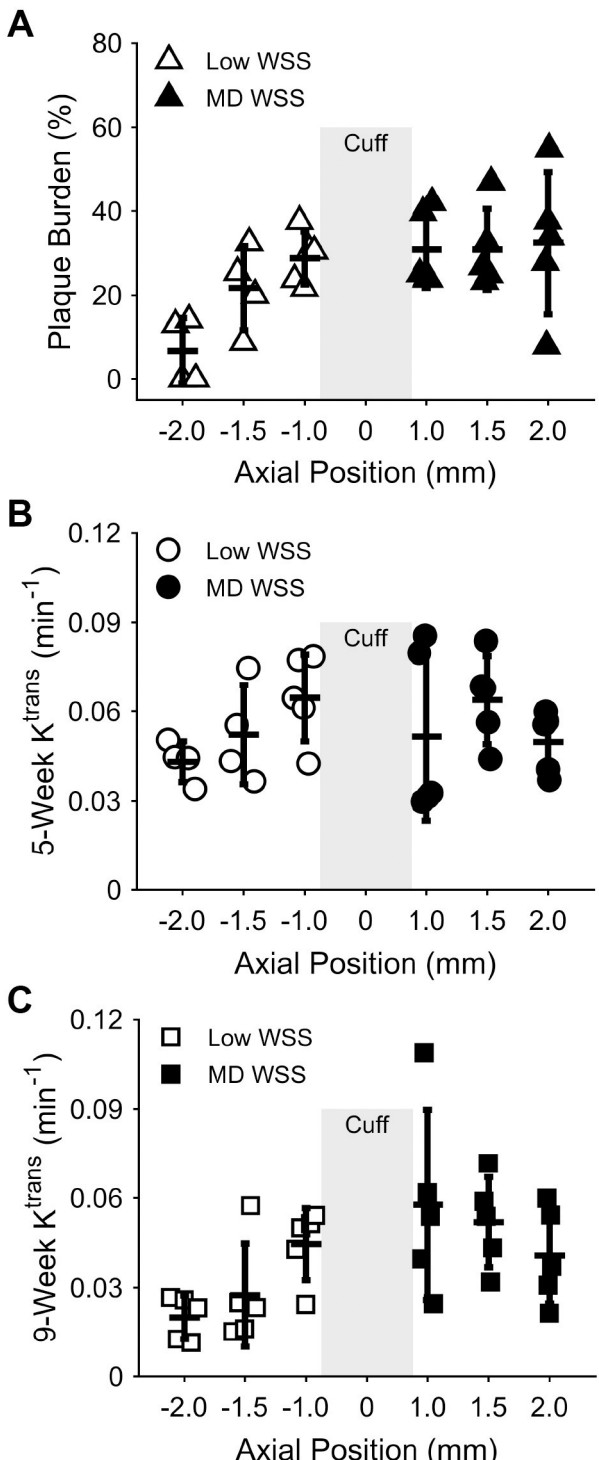

**Fig 3. Plaque burden and NP accumulation over the length of the instrumented artery.** (A) Plaque burden in mice injected with NPs at 9 weeks after cuff placement, (B) NP accumulation assessed via $K^{trans}$ at 5 weeks after cuff placement, and (C) $K^{trans}$ at 9 weeks after cuff placement as a function of position (in mm) from the center of the cuff (no plaque developed in the cuff region and there are no $K^{trans}$ values due to the cuff creating an unreliable signal; thus, no data are presented in this region). Data points represent the mean plaque burden and $K^{trans}$ values for each mouse injected with NPs ($n$ = 5) at each MRI slice location. Bars are mean±SD. These data were not evaluated for statistical differences between vessel regions.

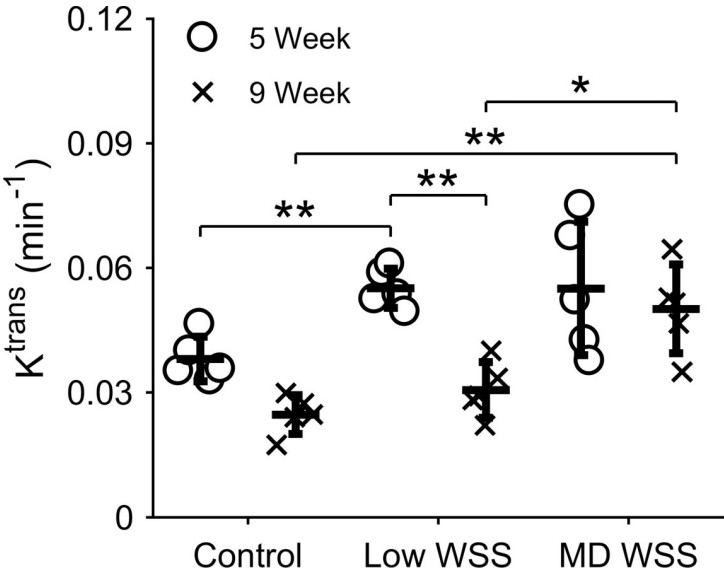

**Fig 4. NP accumulation assessed via K$^{trans}$ in disturbed and normal WSS conditions.** K$^{trans}$ in the low and multidirectional (MD) WSS regions of the instrumented carotid arteries and contralateral control carotid arteries at 5 and 9 weeks after cuff placement. Each data point represents the mean K$^{trans}$ from the three DCE-MRI slices closest to the cuff within each WSS region of the instrumented artery and the mean K$^{trans}$ from the central portion of the control artery for each mouse that was injected with NPs ($n$ = 5). Bars are mean±SD. *$P$<0.05 is considered statistically significant.

We also assessed whether the control carotid arteries in ApoE$^{-/-}$ mice exhibited a higher K$^{trans}$ compared to the carotid arteries of C57BL/6 control mice (e.g., due to the presence of hypercholesterolemia) and found no differences at either time point (**S3 Fig**). C57BL/6 carotid arteries showed a mean K$^{trans}$ value of 0.030±0.0074 min$^{-1}$ compared with 0.038±0.0053 min$^{-1}$ and 0.025±0.0047 min$^{-1}$ in the ApoE$^{-/-}$ control carotid arteries at 5 ($p$ = 0.18) and 9 ($p$ = 0.25) weeks, respectively.

## Differential NP accumulation in plaques is influenced by plaque phenotype

To better understand our finding of changes in K$^{trans}$ from 5 to 9 weeks in the low WSS region, we evaluated changes in the constituents and size of plaques in both regions of the instrumented arteries. In plaques induced by low WSS, α-SMA significantly decreased 0.46-fold (*$p$ = 0.023) and plaque burden significantly increased 1.60-fold (*$p$ = 0.048) from 5 to 9 weeks after cuff placement, while lipid exhibited a non-significant increase of 1.59-fold ($p$ = 0.13) and collagen was unchanged (**Fig 5**). The distribution of α-SMA also changed from nearly uniform at 5 weeks to more localized to the plaque-lumen interface at 9 weeks (**Fig 5I**). Although plaques induced by multidirectional WSS exhibited mostly similar trends, none of the differences in plaque constituents or size reached statistical significance (**Fig 6**).

To evaluate how NP accumulation may be influenced by plaque phenotype, each of the aforementioned constituents and plaque area were correlated to K$^{trans}$ from co-registered DCE-MRI slices at 9 weeks after cuff placement. In plaques induced by low WSS (**Fig 7A and 7B**), the fluorescence intensity of α-SMA exhibited a significant inverse correlation with K$^{trans}$ of -0.87 (*$p$ = 0.014) (**Fig 7C**). This inverse correlation was found to stem from the plaque cap (considered to cover the first 13.3 μm of plaque thickness from the lumen), where separate evaluation revealed an even stronger correlation of -0.92 (**$p$ = 0.0039) (**Fig 7D**). On the other hand, the plaque body (whole plaque minus the cap) in this region showed a non-significant

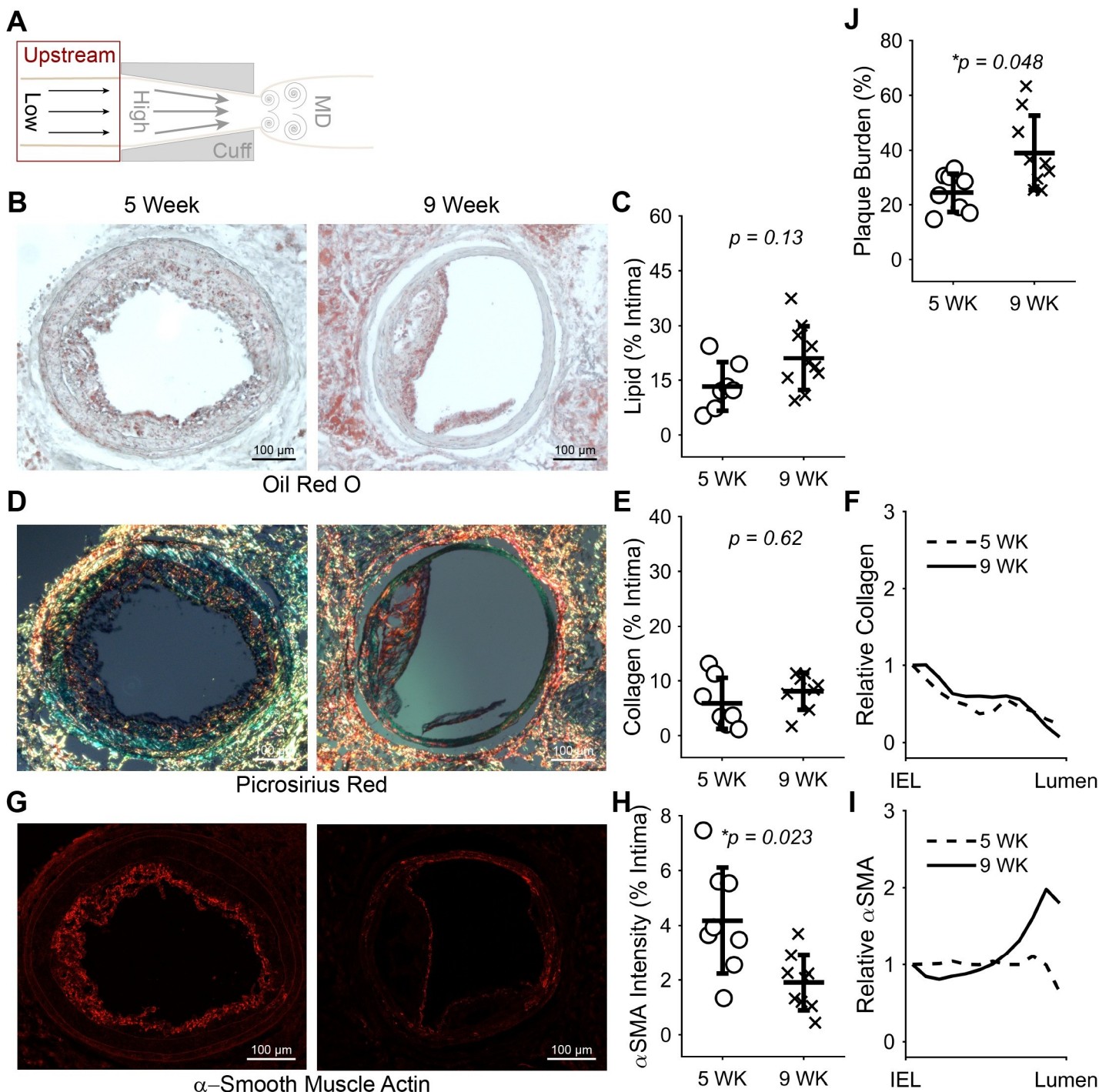

**Fig 5. Constituents and size of plaques induced by low WSS at 5 versus 9 weeks.** (A) Diagram of the instrumented carotid artery highlighting the focus of this figure on the low WSS (upstream) region. (B) Representative histology sections stained with oil red O to detect lipid and (C) plot of mean lipid area normalized by intima area across all mice at 5 ($n$ = 7 mice) and 9 ($n$ = 10) weeks after cuff placement. (D) Representative histology sections stained with picrosirius red to detect collagen, (E) plot of mean collagen area normalized by intima area at 5 ($n$ = 7) and 9 ($n$ = 8) weeks, and (F) plot of mean collagen distribution from the IEL to the lumen normalized by the value at the IEL at 5 and 9 weeks. (G) Representative histology sections stained for α-SMA, (H) plot of mean α-SMA intensity normalized by intima area at 5 ($n$ = 8) and 9 ($n$ = 10) weeks, and (I) plot of mean α-SMA intensity distribution from the IEL to the lumen normalized by the value at the IEL at 5 and 9 weeks. (J) Plot of mean plaque burden at 5 and 9 weeks. In the scatter plots, each data point represents the average of all viable histological sections for the given stain in the plaque region of the upstream vessel segment of one mouse. Bars are mean±SD. *$P$<0.05 is considered statistically significant.

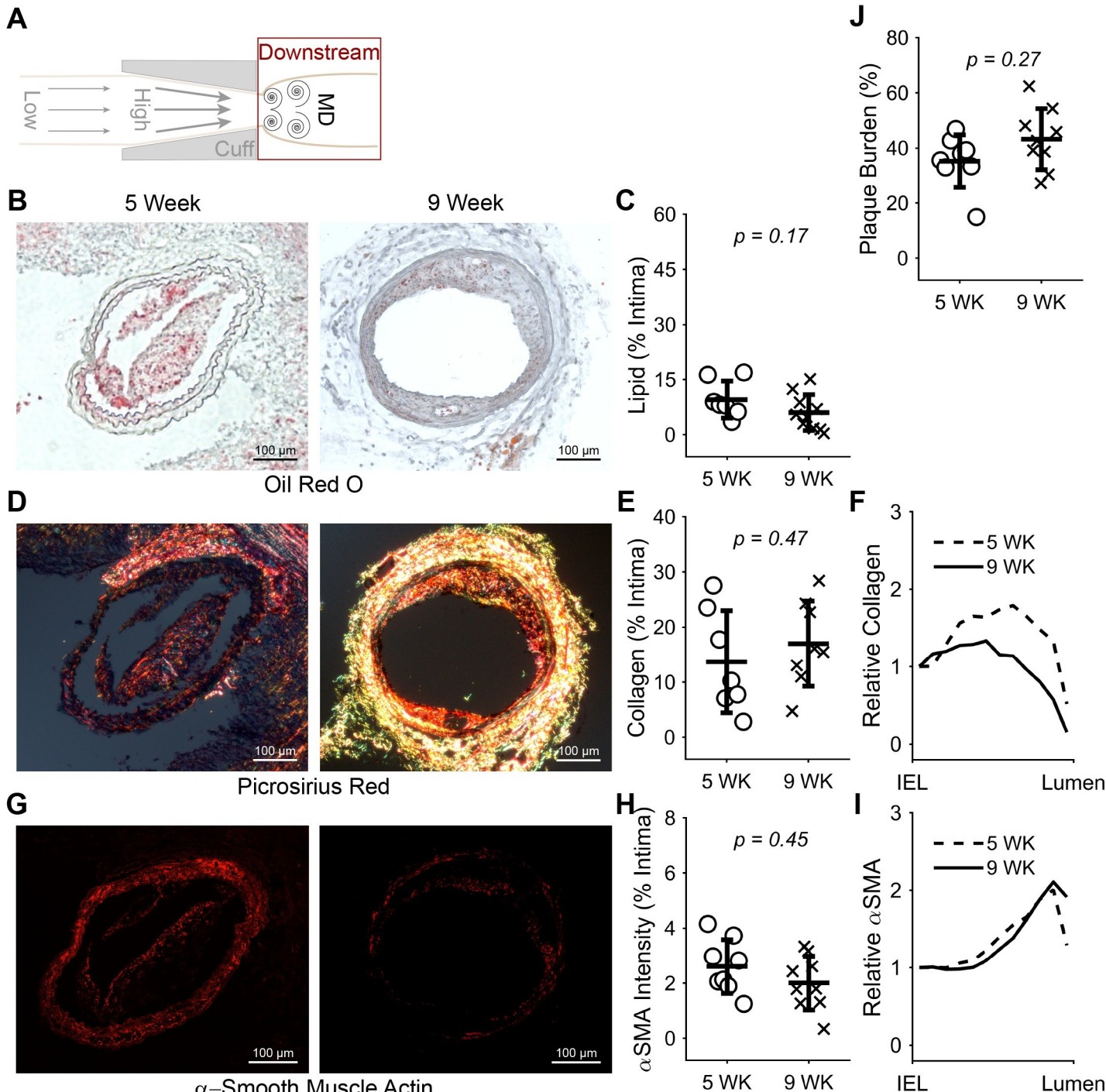

**Fig 6. Constituents and size of plaques induced by multidirectional WSS at 5 versus 9 weeks.** (A) Diagram of the instrumented carotid artery highlighting the focus of this figure on the multidirectional WSS (downstream) region. (B) Representative histology sections stained with oil red O to detect lipid and (C) plot of mean lipid area normalized by intima area across all mice at 5 ($n = 7$ mice) and 9 ($n = 10$) weeks after cuff placement. (D) Representative histology sections stained with picrosirius red to detect collagen, (E) plot of mean collagen area normalized by intima area at 5 ($n = 7$) and 9 ($n = 8$) weeks, and (F) plot of mean collagen distribution from the IEL to the lumen normalized by the value at the IEL at 5 and 9 weeks. (G) Representative histology sections stained for α-SMA, (H) plot of mean α-SMA intensity normalized by intima area at 5 ($n = 8$) and 9 ($n = 10$) weeks, and (I) plot of mean α-SMA intensity distribution from the IEL to the lumen normalized by the value at the IEL at 5 and 9 weeks. (J) Plot of mean plaque burden at 5 and 9 weeks. In the scatter plots, each data point represents the average of all viable histological sections for the given stain in the plaque region of the downstream vessel segment of one mouse. Bars are mean±SD. *$P < 0.05$ is considered statistically significant.

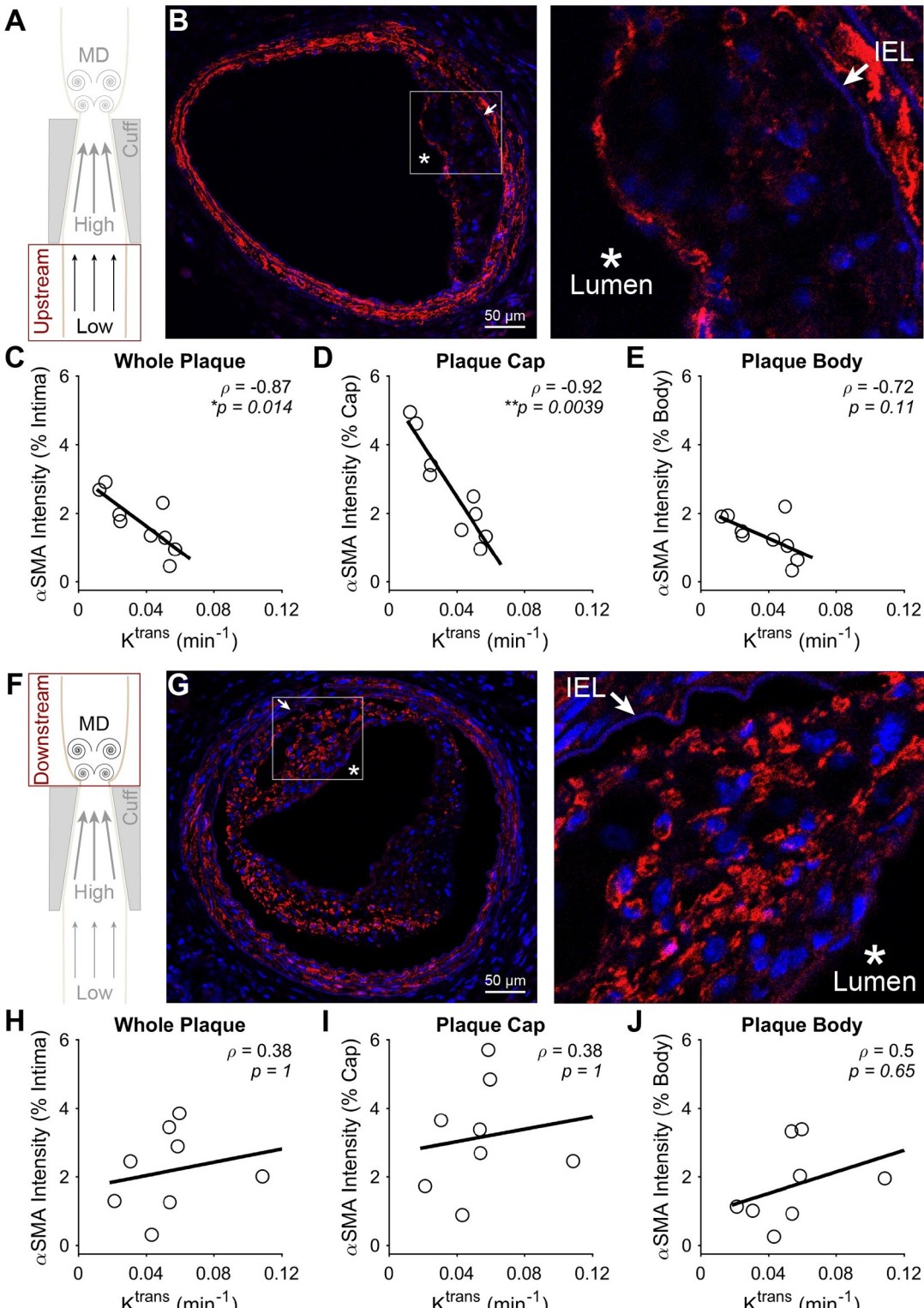

**Fig 7. Correlation between α-SMA and NP accumulation assessed via K^trans at 9 weeks after cuff placement.** (A) Diagram of the instrumented carotid artery highlighting the focus of this part of the figure on the low WSS (upstream) region. (B) A representative histology section and higher magnification insert of the plaque region, where the lumen and IEL are identified. (C-E) The correlation between α-SMA and K$^{trans}$ in different regions of the plaques induced by low WSS, including the (C) entire plaque, (D) plaque cap (13.3 μm), and (E) plaque body minus the cap. (F) Diagram of the instrumented carotid artery

highlighting the focus of this part of the figure on the multidirectional (MD) WSS (downstream) region. (G) A representative histology section and higher magnification insert of the plaque region. (H-J) The correlation between α-SMA and $K^{trans}$ in different regions of the plaques induced by multidirectional WSS, including the (H) entire plaque, (I) plaque cap (13.3 μm), and (J) plaque body minus the cap. Each data point of each plot represents the mean α-SMA fluorescence intensity normalized by intimal area across all histological sections associated with a DCE-MRI slice, from which $K^{trans}$ was obtained (in mice with viable histological sections associated with more than one DCE-MRI slice, more than one pairing was used; all mice injected with NPs ($n = 5$) are represented in all plots). The black line in each plot represents a linear regression of the data to visualize the trend. Spearman's correlation coefficient, $\rho$, and associated $p$-value are also given. $^*P < 0.05$ is considered statistically significant.

correlation of -0.72 ($p = 0.11$) (**Fig 7E**). In contrast to low WSS, α-SMA in plaques induced by multidirectional WSS exhibited no correlation with $K^{trans}$ (**Fig 7F–7J**). Collagen in plaques induced by low WSS exhibited a complementary trend to α-SMA, but with lower absolute correlation coefficients that did not achieve statistical significance. The correlation between $K^{trans}$ and collagen was 0.70 ($p = 0.13$) in the whole plaque and 0.73 ($p = 0.093$) in the plaque body, but exhibited a much lower value in the plaque cap of 0.10 ($p = 1$) (**Fig 8A–8E**). Collagen in plaques induced by multidirectional WSS showed similar results to α-SMA, where none of the plaque components demonstrated marked or significant correlation coefficient values with $K^{trans}$ (**Fig 8F–8J**). Lipid showed no correlation with $K^{trans}$ in either segment of the instrumented vessel (**S4 Fig**). In addition to these plaque constituents, we also evaluated the correlation between plaque area and $K^{trans}$. Plaque area exhibited a strong positive correlation with $K^{trans}$ in plaques induced by low WSS (**Fig 9A–9D**), particularly in the whole plaque (rho = 0.95, $^{**}p = 0.0034$) and body (rho = 0.95, $^{**}p = 0.0034$), but not the cap (rho = 0.71, $p = 0.17$). There was no correlation between plaque area and $K^{trans}$ in plaques induced by multidirectional WSS (**Fig 9E–9H**).

Since there were strong correlations between smooth muscle, collagen, and plaque area in plaques induced by low WSS, but not MD WSS, we also evaluated differences in these plaque characteristics between the flow conditions at 9 weeks after cuff placement. Interestingly, neither plaque burden nor α-SMA were different between these regions (**Fig 10A and 10B**). However, plaques induced by low WSS exhibited a statistically significantly higher amount of lipid (3.51-fold, $^{**}p = 0.0005$) and lower amount of collagen (0.48-fold, $^*p = 0.030$) compared to those induced by multidirectional WSS (**Fig 10C and 10D**). In addition, we evaluated a marker of cell proliferation, Ki67, which was also statistically higher in plaques induced by low WSS versus multidirectional WSS (3.70-fold, $^*p = 0.015$). Separate evaluation in the plaque cap region showed an even larger difference in Ki67 between these two plaque phenotypes of 4.80-fold ($^*p = 0.018$) (**S5 Fig**).

## Discussion

WSS in regions of disturbed blood flow promotes endothelial dysfunction and atherosclerosis, and many studies have demonstrated that specific disturbed blood flow conditions promote certain plaque phenotypes [1, 8–10]. In this study, we evaluated the influence of both the WSS condition and plaque phenotype on the accumulation of NPs within atherosclerotic arteries. At 5 weeks after cuff placement, we found that NP accumulation in the low WSS regions was statistically higher than the contralateral control arteries, which exhibit laminar flow, but there were no differences in accumulation between the two plaque phenotypes at this intermediate stage. However, while NP accumulation was the same from 5 to 9 weeks in the multidirectional WSS region, NP accumulation was statistically lower in the low WSS region at 9 weeks compared to 5 weeks. Direct comparison of the two WSS regions at 9 weeks also demonstrated a statistically lower NP accumulation in the low WSS compared to multidirectional WSS regions. This delayed difference between the two WSS regions suggests that different disturbed

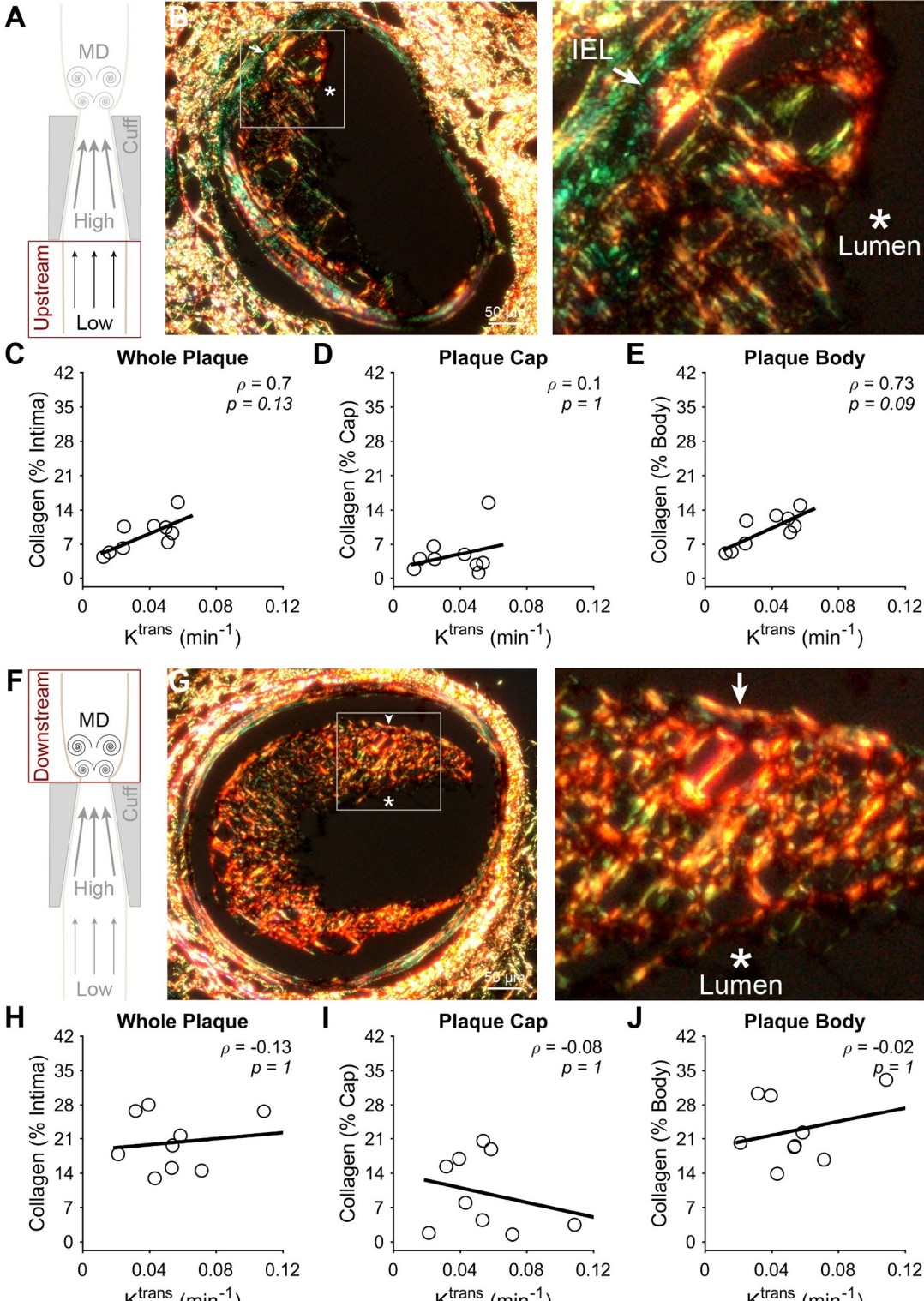

**Fig 8. Correlation between collagen and NP accumulation assessed via K$^{trans}$ at 9 weeks after cuff placement.** (A) Diagram of the instrumented carotid artery highlighting the focus of this part of the figure on the low WSS (upstream) region. (B) A representative histology section and higher magnification insert of the plaque region, where the lumen and IEL are identified. (C-E) The correlation between collagen and K$^{trans}$ in different regions of the plaques induced by low WSS, including the (C) entire plaque, (D) plaque cap (13.3 μm), and (E) plaque body minus the cap. (F) Diagram of the instrumented carotid artery

highlighting the focus of this part of the figure on the multidirectional WSS (downstream) region. (G) A representative histology section and higher magnification insert of the plaque region. (H-J) The correlation between collagen and $K^{trans}$ in different regions of the plaques induced by multidirectional (MD) WSS, including the (H) entire plaque, (I) plaque cap (13.3 µm), and (J) plaque body minus the cap. Each data point of each plot represents the mean collagen area normalized by intimal area across all histological sections associated with a DCE-MRI slice, from which $K^{trans}$ was obtained (in mice with viable histological sections associated with more than one DCE-MRI slice, more than one pairing was used; all mice injected with NPs ($n$ = 5) are represented in all plots). The black line in each plot represents a linear regression of the data to visualize the trend. Spearman's correlation coefficient, $\rho$, and associated $p$-value are also given. *$P<0.05$ is considered statistically significant.

blood flow conditions do not directly affect NP accumulation, at least not for the specific NPs used herein; rather, the effect is indirect in that the final plaque phenotypes induced by the two WSS regions drive differences in NP accumulation.

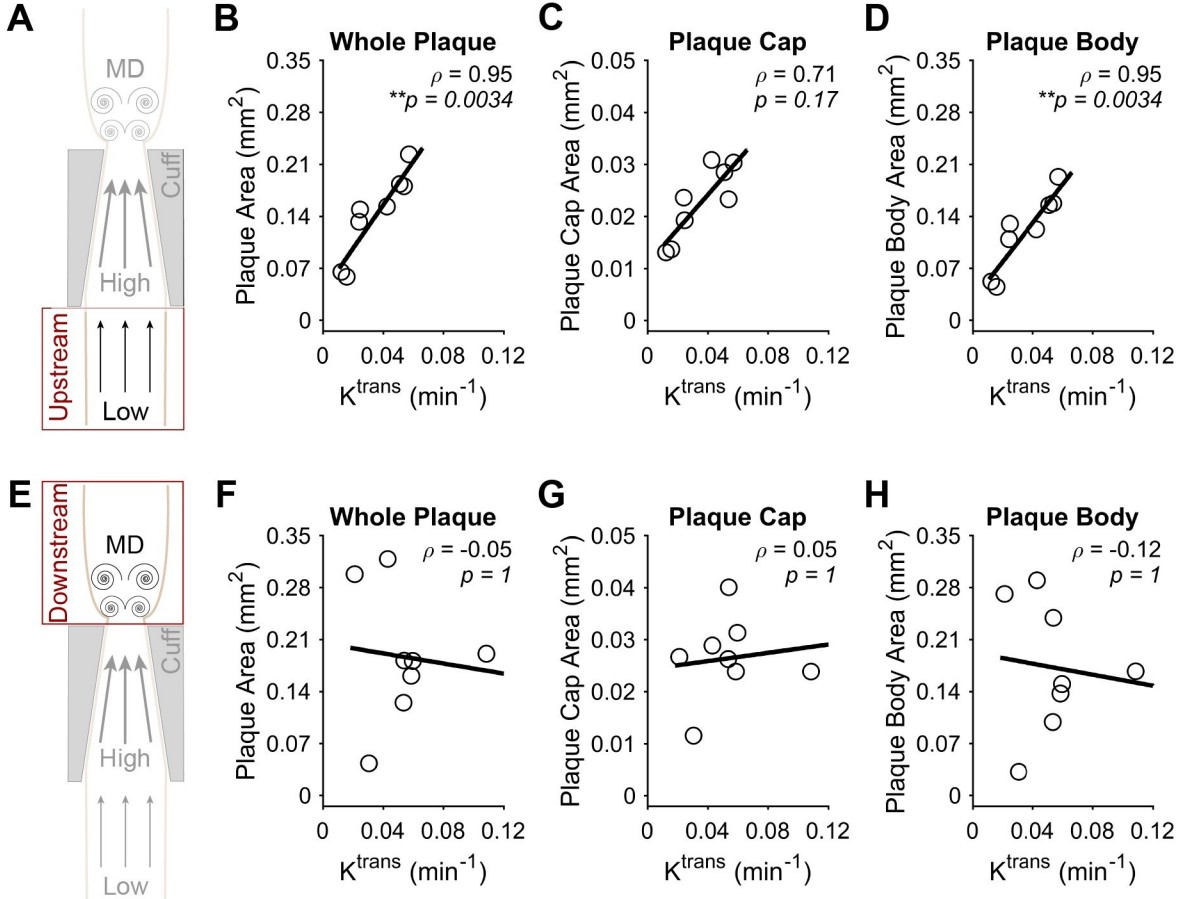

**Fig 9. Correlation between plaque area and NP accumulation assessed via $K^{trans}$ at 9 weeks after cuff placement.** (A) Diagram of the instrumented carotid artery highlighting the focus of this part of the figure on the low WSS (upstream) region. (B-D) The correlation between plaque area and $K^{trans}$ in different regions of the plaques induced by low WSS, including the (B) entire plaque, (C) plaque cap (13.3 µm), and (D) plaque body minus the cap. (E) Diagram of the instrumented carotid artery highlighting the focus of this part of the figure on the multidirectional WSS (downstream) region. (F-H) The correlation between plaque area and $K^{trans}$ in different regions of the plaques induced by multidirectional (MD) WSS, including the (F) entire plaque, (G) plaque cap (13.3 µm), and (H) plaque body minus the cap. Each data point of each plot represents the mean plaque area across all histological sections associated with a DCE-MRI slice, from which $K^{trans}$ was obtained (in mice with viable histological sections associated with more than one DCE-MRI slice, more than one pairing was used; all mice injected with NPs ($n$ = 5) are represented in all plots). The black line in each plot represents a linear regression of the data to visualize the trend. Spearman's correlation coefficient, $\rho$, and associated $p$-value are also given. *$P<0.05$ is considered statistically significant.

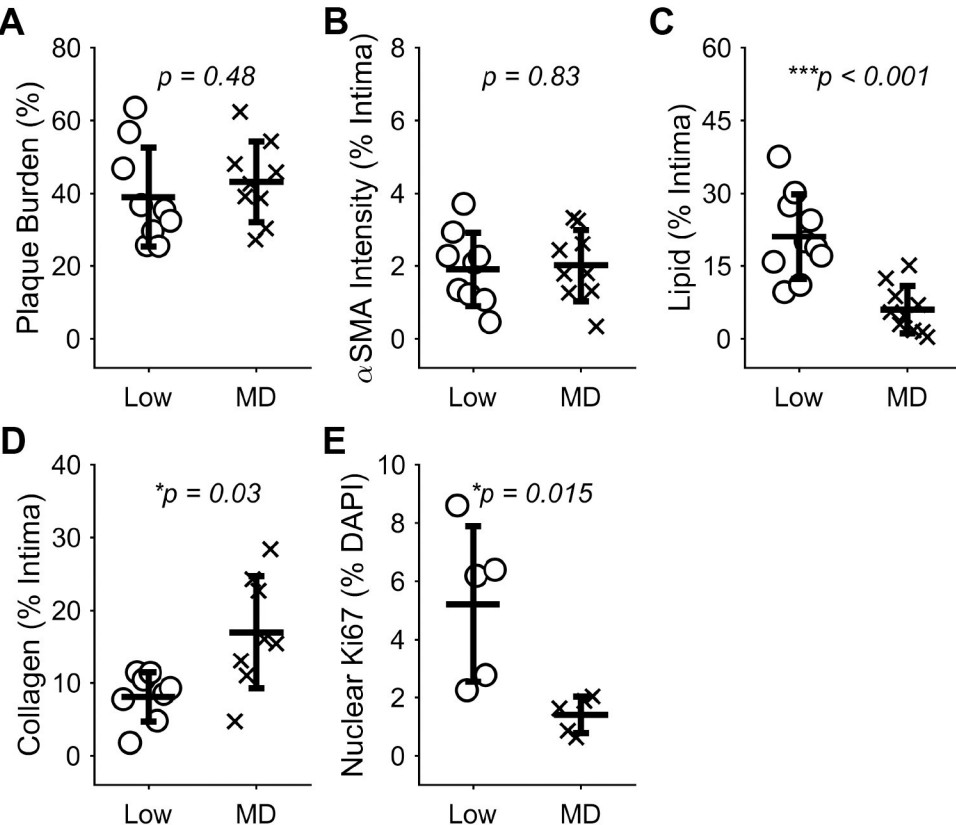

**Fig 10. Characteristics of plaques induced by low WSS compared to multidirectional (MD) WSS.** Plaque size and constituents assessed over the whole plaque, including: (A) Plaque burden ($n = 10$ mice), (B) $\alpha$-SMA intensity ($n = 10$), (C) lipid ($n = 10$), (D) collagen ($n = 8$), and (E) area of Ki67 in the nucleus of cells normalized by total cell nuclear (DAPI) area ($n = 5$). Each data point of each plot represents the average of a given readout across all viable histological sections in each vessel segment of one mouse. Bars are mean±SD. *$P < 0.05$ is considered statistically significant.

Following this observation, we found that the plaque constituent exhibiting the most significant (inverse) correlation with $K^{trans}$ was $\alpha$-SMA at the plaque-lumen interface (i.e., the cap) of plaques induced by low WSS (unstable plaque). Since there was no correlation between collagen and $K^{trans}$ in the cap region of these plaques, our findings suggest that the presence of smooth muscle cells at this lumen interface was the principal determinant of NP transport. There was also no correlation between $\alpha$-SMA or collagen and $K^{trans}$ in plaques induced by multidirectional WSS. Interestingly, we found no difference in the relative amount of $\alpha$-SMA between the two plaque phenotypes, although we did find that $\alpha$-SMA significantly decreased and changed distribution from the IEL to the lumen in the unstable (low WSS) plaques, but not the stable (multidirectional WSS) plaques. In addition, the unstable plaques had higher lipid, lower collagen, and higher Ki67 expression compared to stable plaques, as expected and in line with previous studies that characterized the plaque phenotypes in this mouse model [10, 11, 16]. Together, these findings suggest that smooth muscle cells in the unstable plaques are slightly more distributed towards the lumen and have a more synthetic phenotype compared to those in stable plaques, which may underlie the disruption to NP transport.

In humans, the fibrous cap (of a fibrous cap atheroma) has been described by Virmani et al. [37] as "a distinct layer of connective tissue completely covering the lipid core" that "consists purely of smooth muscle cells in a collagenous proteoglycan matrix." In contrast, pathological intimal thickening is "characterized by the presence of smooth muscle cells interspersed within

extracellular matrix towards the lumen and areas of extracellular lipid accumulation" [38], suggesting lesser organization. Our histological evaluation of the unstable and stable murine plaques herein suggests a similar organization of the smooth muscle to that seen in fibrous cap atheroma and pathological intimal thickening, respectively, in humans that leads to higher transport in more stable plaques. This fits with our observation of higher $K^{trans}$ in stable plaques induced by multidirectional WSS compared to unstable plaques induced by low WSS. It is also supported by our observation of a positive correlation between collagen and $K^{trans}$ in the body of plaques induced by low WSS that suggests the unstable plaques have some variability between them and those that are less advanced (e.g., more collagen) are more permeable to NPs. Plaque area was also found to positively correlate with $K^{trans}$ in plaques induced by low WSS, but this may be simply due to the larger space for NPs to accumulate (although no correlation was found in plaques induced by multidirectional WSS). Larger overall plaque size does not necessarily correspond to a more advanced phenotype [39, 40].

Previous studies in mice have also reported lower transport in unstable versus stable plaques, despite differences in experimental procedures (e.g., NP formulations) from the present work. One study assessed the uptake of 16 nm micelles (similar to the 13 nm NPs used in this work) based on simple signal enhancement in a similar mouse model and showed a trending decrease in unstable plaque uptake with plaque progression, while uptake in the stable plaque did not show a decrease with progression [41]. Another study showed significantly decreased NP (90 nm hyaluronan NP) accumulation in the aortic arch of ApoE$^{-/-}$ mice (where atherosclerosis naturally develops) at 12 weeks compared to 6 weeks on a Western diet [34]. They attributed this decrease to a finding of increased endothelial cell junction continuity that improved barrier function with plaque progression (which may occur due to increased flow on the upstream side of plaques causing significant lumen stenosis) and changes in plaque morphology, wherein advanced plaques demonstrated smooth muscle cell migration into the intima, as found herein. Important to these observations is that NP uptake in mouse atherosclerotic plaques takes place predominantly across disrupted macrovascular endothelium [34]. Though neovascularization into the plaque from the adventitia has been noted in human studies [42, 43], work in mice has shown limited connection between plaque interior and adventitial microvasculature [44]. Other studies in a rabbit model [45] and human patients [46] have shown that extensive neovascularization was associated with plaque progression to a vulnerable phenotype and increased permeability due to the formation of immature and leaky microvessels.

Our observation of differential NP uptake in unstable versus stable murine plaques at the fully developed stage, but not an intermediate stage, of growth, suggests that these NPs, together with DCE-MRI, could be used as a non-invasive diagnostic of plaque phenotype in mouse models of atherosclerosis. Such a diagnostic would benefit studies of plaque progression, as well as those examining the efficacy of novel therapeutics in promoting plaque regression, by eliminating the need for costly and time-consuming histological evaluation of the plaque at intermediate time points. Further characterization and development of this approach could also lead to a non-invasive diagnostic for patients to better characterize plaque stage. Future work will seek to improve the efficacy of this diagnostic by using NPs targeted to the dysfunctional endothelium (e.g., VCAM-1 [12]) or plaque components (e.g., CD146 on foam cells [47]) as well as identifying how NP size affects their accumulation across various stages of plaque development. While elegant work has been conducted comparing NP size in atherosclerosis [48], it was done in the context of improving therapy by mimicking high density lipoprotein at a single stage of plaque development. Tissue fluorescence-based comparisons between uptake in the aortic arch were performed, but was not a focus of the study, so direct quantitative size-accumulation comparisons were not made; however, smaller NPs showed

higher aorta-to-spleen accumulation ratios as compared to larger NPs. The reduction in accumulation in larger NPs is likely a result of the level of disruption of the macrovascular endothelium at the advanced stage of plaque development in their model (ApoE$^{-/-}$ mice fed a high-fat diet, 42% calories from fat, for 16 weeks).

## Conclusion

NPs may offer an impactful strategy for the development of a novel diagnostic of atherosclerotic plaque phenotype. To accomplish this goal, there is a need for better understanding of how the blood flow environment and plaque characteristics affect NP accumulation and retention. Using a mouse model of atherosclerosis where one carotid artery was instrumented with a constrictive cuff to induce low WSS upstream of the cuff and multidirectional WSS downstream of the cuff, we found that accumulation and retention kinetics of folic acid-coated gadolinium (FA-Gd) NPs were not directly influenced by the WSS condition, but rather, the plaque phenotype that develops in each WSS condition. In particular, the plaques in both regions at 5 weeks showed the same NP accumulation, but those induced by low WSS at 9 weeks exhibited lower NP accumulation than plaques induced by multidirectional WSS. Correlations of NP accumulation to the features of plaques induced by low WSS revealed that this difference may be due to the presence of smooth muscle cells at the plaque-lumen interface (i.e., the cap). Overall, this result demonstrates the ability of NPs to identify different plaque phenotypes based on differences in passive accumulation. Future work will investigate NP-based diagnostics that employ active targeting strategies of the dysfunctional endothelium and plaque components to maximize accumulation and retention within different plaque types.

## Supporting information

**S1 Fig. T1-weighted images of carotid arteries over time before and after injection of nanoparticles.**
(TIF)

**S2 Fig. Gd-time curves in the cuffed and control arteries.** Curves represent a running average of 5 time points taken from a single slice in the relevant vessel region from a single animal.
(TIF)

**S3 Fig. Additional comparisons of K$^{trans}$.** (A) No statistical differences were seen in Ktrans from C57BL/6 mice (four vessels from $n = 2$ mice) versus control arteries from ApoE mice ($n = 5$ mice) at 5 and 9 weeks. (B) Four additional mice used for histological evaluation of plaque features at 5 weeks after cuff placement were also injected with NPs immediately prior to culling. No statistical differences in K$^{trans}$ were seen in this group of mice (Group 2) versus the five mice injected with NPs at 5 and 9 weeks after cuff placement (Group 1). Each data point represents the mean K$^{trans}$ from the central portion of the control artery or the three DCE-MRI slices closest to the cuff within each WSS region of the instrumented artery for each mouse that was injected with NPs. Bars are mean±SD.
(TIF)

**S4 Fig. Correlation between lipid and NP accumulation assessed via K$^{trans}$ at 9 weeks after cuff placement.** (A) Diagram of the instrumented carotid artery highlighting the focus of this part of the figure on the low WSS (upstream) region. (B-D) The correlation between lipid and K$^{trans}$ in different regions of plaques induced by low WSS, including the (B) entire plaque, (C) plaque cap (13.3 μm), and (D) plaque body minus the cap. (E) Diagram of the instrumented carotid artery highlighting the focus of this part of the figure on the multidirectional (MD)

WSS (downstream) region. (F-H) The correlation between lipid and $K^{trans}$ in different regions of plaques induced by multidirectional WSS, including the (F) entire plaque, (G) plaque cap (13.3 μm), and (H) plaque body minus the cap. Each data point of each plot represents the mean lipid (normalized by intima area) across all histological sections associated with a DCE-MRI slice, from which $K^{trans}$ was obtained (in mice with viable histological sections associated with more than one DCE-MRI slice, more than one pairing was used; all mice injected with NPs (*n* = 5) are represented in all plots). The black line in each plot represents a linear regression of the data to visualize the trend. Spearman's correlation coefficient, *ρ*, and associated *p*-value are also given. *$P<0.05$ is considered statistically significant.
(TIF)

**S5 Fig. Comparison of Ki67 staining in plaques induced by low versus multidirectional WSS.** Representative histology sections of Ki67 counterstained with DAPI in plaques induced by (A) low and (B) multidirectional WSS. Plots of nuclear Ki67 area as a percentage of total DAPI area in the (C) whole plaque and (D) plaque cap across all mice evaluated (*n* = 5). Each data point of each plot represents the average nuclear Ki67 (% DAPI) across all viable histological sections in each vessel segment of one mouse. Bars are mean±SD. *$P<0.05$ is considered statistically significant.
(TIF)

## Acknowledgments

We thank veterinarians Dr. Craig Kreikemeier-Bower and Dr. Anna Fitzwater for their assistance in caring for the mice and performing instrumentation of the left carotid arteries.

## Author Contributions

**Conceptualization:** Yiannis S. Chatzizisis, Forrest M. Kievit, Ryan M. Pedrigi.

**Data curation:** Hunter A. Miller, Morgan A. Schake, Evan T. Curtis, Connor C. Gee, Ian S. McCue, Thomas J. Ripperda, Jr.

**Formal analysis:** Hunter A. Miller, Morgan A. Schake, Forrest M. Kievit, Ryan M. Pedrigi.

**Funding acquisition:** Forrest M. Kievit, Ryan M. Pedrigi.

**Investigation:** Hunter A. Miller, Morgan A. Schake, Evan T. Curtis, Connor C. Gee, Ian S. McCue, Thomas J. Ripperda, Jr.

**Methodology:** Hunter A. Miller, Morgan A. Schake, Badrul Alam Bony, Evan T. Curtis, Ian S. McCue, Forrest M. Kievit, Ryan M. Pedrigi.

**Project administration:** Forrest M. Kievit, Ryan M. Pedrigi.

**Resources:** Hunter A. Miller, Morgan A. Schake, Badrul Alam Bony, Evan T. Curtis, Forrest M. Kievit, Ryan M. Pedrigi.

**Software:** Morgan A. Schake.

**Supervision:** Forrest M. Kievit, Ryan M. Pedrigi.

**Validation:** Hunter A. Miller, Morgan A. Schake, Forrest M. Kievit, Ryan M. Pedrigi.

**Visualization:** Hunter A. Miller, Morgan A. Schake, Forrest M. Kievit, Ryan M. Pedrigi.

**Writing – original draft:** Hunter A. Miller, Morgan A. Schake, Forrest M. Kievit, Ryan M. Pedrigi.

**Writing – review & editing:** Hunter A. Miller, Morgan A. Schake, Badrul Alam Bony, Evan T. Curtis, Connor C. Gee, Ian S. McCue, Thomas J. Ripperda, Jr., Yiannis S. Chatzizisis, Forrest M. Kievit, Ryan M. Pedrigi.

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
