## [Decision Letter · Decision Letter 0]

10 May 2021

PONE-D-21-12791

Smooth muscle cells affect differential nanoparticle accumulation in disturbed blood flow-induced murine atherosclerosis

PLOS ONE

Dear Dr. Pedrigi,

Thank you for submitting your manuscript to PLOS ONE. After careful consideration, we feel that it has merit but does not fully meet PLOS ONE’s publication criteria as it currently stands. Therefore, we invite you to submit a revised version of the manuscript that addresses the points raised during the review process.

We look forward to receiving your revised manuscript.

Kind regards,

Vahid Serpooshan, PhD

Academic Editor

PLOS ONE

Journal Requirements:

Reviewers' comments:

Reviewer's Responses to Questions

**Comments to the Author**

1. Is the manuscript technically sound, and do the data support the conclusions?

Reviewer #1: Partly

Reviewer #2: Yes

2. Has the statistical analysis been performed appropriately and rigorously? 

Reviewer #1: Yes

Reviewer #2: Yes

3. Have the authors made all data underlying the findings in their manuscript fully available?

Reviewer #1: Yes

Reviewer #2: Yes

4. Is the manuscript presented in an intelligible fashion and written in standard English?

Reviewer #1: Yes

Reviewer #2: Yes

5. Review Comments to the Author

Reviewer #1: Comment 1: It is not immediately clear in the manuscript why tracking NPs is important. The paper would strongly benefit from including a section in the Introduction and Discussion regarding the importance of NP tracking and what it could be used for, or be a stand-in for.

Comment 2: The authors comment on endothelial cell role in plaque formation and development, but show no data regarding ECs, instead focusing their analysis on SMCs. It is distracting in the Discussion, so I would suggest that they either expand their work to include some EC characterization (staining, PCR, etc), so remove the the EC references from the paper.

Comment 3: The authors only use one marker to stain the smooth muscle cells (aSMA). While this is enough to determine that the cells are likely smooth muscle, I would suggest them to include markers for cytoskeleton (ex: a-Tub) and cell division (ex: Ki67), which would be much more informative regarding the state and role of smooth muscle in the presented study.

Comment 4: Considering that the paper look at quantification of aSMA, I would suggest at least some rounds of qPCR experiments and western blots/ELISA to generate a more robust quantification of the aSMA expression, in addition to fluorescent staining.

Comment 5: The paper would benefit from higher magnification histological images, to better show if there are aSMA changes in expression and alignment within the cells. Additionally, figures 5 and 6 could benefit from higher quality images, as the presented one are hard to read and do not convey the stated results well.

Comment 6: The aSMA plots in figure 5 and 6 are too busy as presented. Consider revising them to highlight the trendline to make them easier to follow.

Comment 7: In Figure 7, the 5 week MD WSS data plot shows points that would suggest an increasing trendline, but the graphed one is shown to be decreasing. Please elaborate if that is correct.

Reviewer #2: The authors investigate the influence of blood flow on the NP trafficking into mouse atherosclerotic lesions. They use a carotid cuff model inducing both low and turbulent blood flow, resulting in the formation of two different plaque phenotypes, ‘so-called’ stable and unstable. They probe the carotid vessel wall noninvasively with dynamic-contrast enhanced MRI at two timepoints, 5 and 9 weeks after cuff placement, using Ktrans as a quantifier. At earlier timepoint, they observe similar enhancement profile at both locations. They find a significant drop in Ktrans at 9 weeks for plaques at low flow region, while no change for plaques under turbulent flow. The authors attribute this drop to the formation of fibrous cup in plaques at low flow region, substantiated by inverse correlation between smooth muscle content and Ktrans. They highlight different distribution of smooth muscle cells in the investigated lesions. The overall conclusion is that the plaque phenotype is the primary influencing factor.

The leading scientific question is relevant but also complex as the blood flow shapes the plaque morphology and indirectly and/or directly nanoparticle accumulation. The methods are well and extensively explained. Manuscript is well written. However, there are several questions:

1) Currently, the reader does not know what is the characteristics of (immature) plaques at 5-week timepoint. Is the size and morphology similar at both locations? Since the aspect of both the blood flow and plaque properties are equally important, this should be addressed.

2) Collagen fraction/distribution is another important parameter in the context of authors’ question. Also, the authors do not report the plaque volume or its average area based on histology, only the plaque burden at 9 weeks based on MRI. These parameters are potentially also relevant and should be investigated/correlated with mean Ktrans.

3) The authors use Ktrans as a noninvasive quantifier of NP trafficking into the plaque. The current manuscripts lacks raw data such as T1w image-time series, average and/or representative contrast enhancement-time curves, which would help the reader to judge the data quality. This is important since the used Tofts model was originally proposed for a low-molecular weight contrast agent.

4) When comparing the authors’ findings to other studies on NPs, the authors should also consider the aspect of NP size. Their nanomaterial is pretty small, which can influence the accumulation pathways.

6. PLOS authors have the option to publish the peer review history of their article (what does this mean?). If published, this will include your full peer review and any attached files.

Reviewer #1: No

Reviewer #2: No

---

## [Author Response · Author response to Decision Letter 0]

17 Oct 2021

We thank the reviewers for their careful review of our manuscript and the important points that they raised. We have amended our manuscript in line with the reviewers’ suggestions, which has resulted in a greatly improved manuscript. Most importantly, we:

• Improved four of the original figures, added four figures to the main paper, and added five figures to supplemental material. 

• Improved the Introduction to better describe why tracking NPs is important in the development of better diagnostics of plaque phenotype. We decided to focus the Introduction (and Discussion) on diagnostics and, so, removed the points about therapeutic potential.

• Added 13 mice to characterize differences in plaque constituents and size at 5 (intermediate) versus 9 (fully-developed) weeks after cuff placement. 

• Added histological evaluation of collagen and Ki67.

• Improved the images and plots of the figures showing correlations between aSMA and Ktrans and added additional figures for correlations between Ktrans and collagen, as well as Ktrans and plaque size.

• Added a figure showing differences between plaques induced by low versus multidirectional WSS to corroborate previous work showing that these are different plaque phenotypes and support a rationale for the differences seen in NP accumulation (Ktrans) at 9 weeks. 

• Changed the correlation approach from Pearson’s (parametric) to Spearman’s (non-parametric) and included a Bonferonni correction to these correlation plots (implemented as a correction for comparing three plaque locations of each plaque constituent), which changed the previous p-values (the overarching message and conclusions of the paper did not change). 

A response to each reviewer concern is given below. 

Reviewer #1: 

Comment 1: It is not immediately clear in the manuscript why tracking NPs is important. The paper would strongly benefit from including a section in the Introduction and Discussion regarding the importance of NP tracking and what it could be used for, or be a stand-in for.

Reply: We thank the reviewer for the comment. To address this concern, we have rewritten the first paragraph of the Introduction to better describe the limitations of current diagnostics of plaque phenotype and how tracking NPs can help to overcome those limitations. Namely, NPs can be tracked in atherosclerotic arteries to relate accumulation in plaques to plaque phenotype and changes with progression. We have also removed discussion of how NPs can be used to develop novel therapeutics to focus the motivation of the paper solely on diagnostics. We also added to the last paragraph of the Discussion to further highlight how NP tracking can be used to benefit studies focused on mechanisms of plaque progression and regression. Finally, we make the point that further development of these contrast enhancing NPs could eventually lead to improved non-invasive tracking in patients.

Comment 2: The authors comment on endothelial cell role in plaque formation and development, but show no data regarding ECs, instead focusing their analysis on SMCs. It is distracting in the Discussion, so I would suggest that they either expand their work to include some EC characterization (staining, PCR, etc), so remove the the EC references from the paper.

Reply: We agree with the reviewer comment and have removed the sentences in the Discussion related to the role of endothelial cells in determining plaque phenotype. 

Comment 3: The authors only use one marker to stain the smooth muscle cells (aSMA). While this is enough to determine that the cells are likely smooth muscle, I would suggest them to include markers for cytoskeleton (ex: a-Tub) and cell division (ex: Ki67), which would be much more informative regarding the state and role of smooth muscle in the presented study.

Reply: We agree with the reviewer and have now added histological evaluation of Ki67 to the manuscript (Fig 10E and S5 Figure). We found that it increased in plaques induced by low WSS (upstream vessel segment) compared to those induced by multidirectional WSS (downstream). This finding suggests a more synthetic phenotype of the SMCs in the upstream plaques that may help to explain differences in NP accumulation between the two plaque phenotypes. The Results and Discussion were updated accordingly. This has added considerably to the paper and we thank the reviewer for the suggestion. 

Comment 4: Considering that the paper look at quantification of aSMA, I would suggest at least some rounds of qPCR experiments and western blots/ELISA to generate a more robust quantification of the aSMA expression, in addition to fluorescent staining.

Reply: We appreciate the reviewer’s suggestion, but feel that the approach we used herein is able to quantify aSMA at the level of accuracy necessary for determining relative differences between the two timepoints (5 and 9 weeks) and for correlations with ktrans. Our serial histology approach, which involves the collection, staining, imaging, manual segmentation, and analysis with custom software of a multitude of cryosections over the length of the instrumented artery of each mouse for each stain, including aSMA, represents a robust data set. It is also the only way to assess regional variations in aSMA expression within the plaques, which we have now further elaborated on in the revised manuscript (see Figs. 5I and 6I). To address other concerns raised by the reviewers, we have added 13 mice for the analysis of aSMA at 5 versus 9 weeks and additional stains, including ki67 and collagen, to look at differences between the two plaque regions. Overall, we feel that these additional results combined with those of our original submission, including the NP and DCE-MRI work, represent a substantial amount of work that more than sufficiently supports the overarching message of the paper. 

Comment 5: The paper would benefit from higher magnification histological images, to better show if there are aSMA changes in expression and alignment within the cells. Additionally, figures 5 and 6 could benefit from higher quality images, as the presented one are hard to read and do not convey the stated results well.

Reply: We thank the reviewer for the suggestion. We have changed the presentation of the histology images in Figures 5 and 6 to show the vessel section at a larger size and included an inset of each image at higher magnification.

Comment 6: The aSMA plots in figure 5 and 6 are too busy as presented. Consider revising them to highlight the trendline to make them easier to follow.

Reply: We have decreased the size of the data points and increased the thickness of the trend line in each plot to make them easier to follow. To further simplify the figures and better emphasize the strong correlations with the 9-week Ktrans data, we have also removed the correlations of alpha-SMA with the 5-week Ktrans data. This allowed us to put the low WSS (upstream) and multidirectional WSS (downstream) correlations in one figure for better comparison. 

Comment 7: In Figure 7, the 5 week MD WSS data plot shows points that would suggest an increasing trendline, but the graphed one is shown to be decreasing. Please elaborate if that is correct.

Reply: We have now removed all correlations with the 5-week Ktrans data and focused this part of our results on correlations with the 9-week Ktrans data. 

Reviewer #2: The authors investigate the influence of blood flow on the NP trafficking into mouse atherosclerotic lesions. They use a carotid cuff model inducing both low and turbulent blood flow, resulting in the formation of two different plaque phenotypes, ‘so-called’ stable and unstable. They probe the carotid vessel wall noninvasively with dynamic-contrast enhanced MRI at two timepoints, 5 and 9 weeks after cuff placement, using Ktrans as a quantifier. At earlier timepoint, they observe similar enhancement profile at both locations. They find a significant drop in Ktrans at 9 weeks for plaques at low flow region, while no change for plaques under turbulent flow. The authors attribute this drop to the formation of fibrous cup in plaques at low flow region, substantiated by inverse correlation between smooth muscle content and Ktrans. They highlight different distribution of smooth muscle cells in the investigated lesions. The overall conclusion is that the plaque phenotype is the primary influencing factor.

The leading scientific question is relevant but also complex as the blood flow shapes the plaque morphology and indirectly and/or directly nanoparticle accumulation. The methods are well and extensively explained. Manuscript is well written. However, there are several questions:

1) Currently, the reader does not know what is the characteristics of (immature) plaques at 5-week timepoint. Is the size and morphology similar at both locations? Since the aspect of both the blood flow and plaque properties are equally important, this should be addressed.

Reply: We have now added 13 mice to the study (for a total of 18 mice) to histologically characterize differences in the plaques at 5 vs 9 weeks, including lipid, collagen, aSMA, and plaque burden. We found a significant increase in plaque burden and significant decrease in aSMA in the unstable (upstream) plaque region. These trends were similar in the stable (downstream) plaque region, but not significant. We also evaluated aSMA expression over the thickness of the plaques at 5 vs 9 weeks and found an interesting difference in the unstable plaque region, but not the stable (see new Figs. 5 and 6). This has added substantially to the manuscript and we thank the reviewer for the suggestion. 

2) Collagen fraction/distribution is another important parameter in the context of authors’ question. Also, the authors do not report the plaque volume or its average area based on histology, only the plaque burden at 9 weeks based on MRI. These parameters are potentially also relevant and should be investigated/correlated with mean Ktrans.

Reply: We have now added histological evaluation of collagen (picrosirius red) to the paper. As described in Comment 1, we assessed differences in this and other constituents at 5 and 9 weeks, as well as correlation with Ktrans. We also evaluated the correlation of (average) plaque area with Ktrans. Plaque area was chosen as it was the only metric of plaque size that could be quantified in different parts of the plaque (e.g., cap and body). Collagen showed fairly high correlation values, but they were not statistically significant, whereas plaque area showed significant positive correlations with Ktrans (Figs. 8 and 9). We have updated the Results and Discussion sections of our paper to reflect these new findings. We thank the reviewer for the suggestion. 

3) The authors use Ktrans as a noninvasive quantifier of NP trafficking into the plaque. The current manuscripts lacks raw data such as T1w image-time series, average and/or representative contrast enhancement-time curves, which would help the reader to judge the data quality. This is important since the used Tofts model was originally proposed for a low-molecular weight contrast agent.

Reply: We have added some of the raw T1w images and time curves in the supplementary information to allow the reader to judge data quality (Figs. S1 and S2). Furthermore, we have added additional detail to the methods section to clarify the two-compartment pharmacokinetic model we use is the Patlak model, which has been shown to work with larger contrast agents.

4) When comparing the authors’ findings to other studies on NPs, the authors should also consider the aspect of NP size. Their nanomaterial is pretty small, which can influence the accumulation pathways.

Reply: We have added additional information in the Discussion on the possible role of NP size on accumulation into the plaque where we compare our 13nm NPs to previous studies using similar sized (16nm) and larger (90nm) NPs. We also added to the Discussion by describing future work needs including direct comparisons of NP size on accumulation mechanism similar to, but in more depth than, a previous study comparing NP size in the context of treatment efficacy by mimicking HDL.

---

## [Decision Letter · Decision Letter 1]

15 Nov 2021

Smooth muscle cells affect differential nanoparticle accumulation in disturbed blood flow-induced murine atherosclerosis

PONE-D-21-12791R1

Dear Dr. Pedrigi,

We’re pleased to inform you that your manuscript has been judged scientifically suitable for publication and will be formally accepted for publication once it meets all outstanding technical requirements.

Kind regards,

Vahid Serpooshan, PhD

Academic Editor

PLOS ONE

Additional Editor Comments (optional):

Reviewers' comments:

Reviewer's Responses to Questions

**Comments to the Author**

1. If the authors have adequately addressed your comments raised in a previous round of review and you feel that this manuscript is now acceptable for publication, you may indicate that here to bypass the “Comments to the Author” section, enter your conflict of interest statement in the “Confidential to Editor” section, and submit your "Accept" recommendation.

Reviewer #1: All comments have been addressed

Reviewer #2: All comments have been addressed

2. Is the manuscript technically sound, and do the data support the conclusions?

Reviewer #1: Yes

Reviewer #2: Yes

3. Has the statistical analysis been performed appropriately and rigorously? 

Reviewer #1: Yes

Reviewer #2: Yes

4. Have the authors made all data underlying the findings in their manuscript fully available?

Reviewer #1: Yes

Reviewer #2: Yes

5. Is the manuscript presented in an intelligible fashion and written in standard English?

Reviewer #1: Yes

Reviewer #2: Yes

6. Review Comments to the Author

Reviewer #1: The authors answered all the comments and concerns raised by the reviewers adequately and in good faith. They also added several experiments that were requested, which in my opinion makes this a much stronger submission and a successful publication.

Reviewer #2: The revisions improved the paper considerably, the added data gave a better picture of the studied plaque phenotypes

7. PLOS authors have the option to publish the peer review history of their article (what does this mean?). If published, this will include your full peer review and any attached files.

Reviewer #1: No

Reviewer #2: No

---

## [Editor Report · Acceptance letter]

1 Dec 2021

PONE-D-21-12791R1 

Smooth muscle cells affect differential nanoparticle accumulation in disturbed blood flow-induced murine atherosclerosis 

Dear Dr. Pedrigi:

I'm pleased to inform you that your manuscript has been deemed suitable for publication in PLOS ONE. Congratulations! Your manuscript is now with our production department. 

Kind regards, 

on behalf of

Dr. Vahid Serpooshan 

Academic Editor

PLOS ONE